# Nuclear Pore Proteins in Regulation of Chromatin State

**DOI:** 10.3390/cells8111414

**Published:** 2019-11-09

**Authors:** Terra M. Kuhn, Maya Capelson

**Affiliations:** Department of Cell and Developmental Biology, Penn Epigenetics Institute, Perelman School of Medicine, University of Pennsylvania, Philadelphia, PA 19104, USA; terrak@pennmedicine.upenn.edu

**Keywords:** nuclear pore, nucleoporin, chromatin, chromatin remodeling, histone modifications, chromatin compaction, gene expression, transcription, nuclear organization

## Abstract

Nuclear pore complexes (NPCs) are canonically known to regulate nucleocytoplasmic transport. However, research efforts over the last decade have demonstrated that NPCs and their constituent nucleoporins (Nups) also interact with the genome and perform important roles in regulation of gene expression. It has become increasingly clear that many Nups execute these roles specifically through regulation of chromatin state, whether through interactions with histone modifiers and downstream changes in post-translational histone modifications, or through relationships with chromatin-remodeling proteins that can result in physical changes in nucleosome occupancy and chromatin compaction. This review focuses on these findings, highlighting the functional connection between NPCs/Nups and regulation of chromatin structure, and how this connection can manifest in regulation of transcription.

## 1. Introduction

Nuclear pore complexes (NPCs), composed of ~30 distinct Nucleoporins (Nups), are the main mode of nucleocytoplasmic transport in eukaryotic cells. Since early characterization of NPCs via electron microscopy (EM) in the late 1950s, “intranuclear channels” could be seen extending from nuclear pores into the nucleoplasm [1]. These channels were “cylindrical” in nature and “sharply outlined” in contrast to the more dense adjacent material abutting the nuclear envelope (material, which we now know to be lamina-associated condensed heterochromatin), leading to the idea that such nuclear pore-associated channels may correspond to areas of less dense chromatin. Understanding of chromatin structure and function developed throughout subsequent years, including the finding that active genes were particularly sensitive to DNase digestion [2], facilitating the idea that active genes have an “open” nature to them. In 1985, the initial EM observations of what appeared to be open chromatin at nuclear pores were supported biochemically through experiments showing that accessible, DNase sensitive chromatin specifically localized to the aforementioned nuclear pore-associated intranuclear channels [3]. In the same month, Gunter Blobel published his famous gene-gating hypothesis, predicting decades worth of future research by envisioning the nuclear pore as a scaffold to organize and facilitate transcription, processing of active genes, and efficient export of newly synthesized gene products [4]. The many studies that have since come out linking NPCs and their constituent Nups to not only transcriptional processes, but also to DNA damage repair, telomere maintenance, and mitotis, have been extensively described in previous reviews [5,6,7,8,9,10]. What we focus on in this review is the mounting evidence demonstrating the role of Nups and NPCs specifically in facilitating the upstream step of regulating chromatin state—important for accurate function of downstream transcriptional processes. The early EM observations and DNAse digestion studies described above explicitly suggested a possible functional relationship between nuclear pores and open chromatin states, and studies over the following decades have further supported this notion. Interestingly, interactions between Nups and the genome have been shown to also occur “off-pore”, taking place in the nuclear interior in metazoan cells [11,12,13,14]. This is likely due to the dynamic behavior of certain Nups, which have been shown to have low residence times at nuclear pores and to have intranuclear presence [15,16]. The existence of both on-pore and off-pore binding sites of Nups extends the functional reach of Nups in relation to chromatin structure, suggesting that the initially proposed connection between nuclear pores and chromatin regulation may be extrapolated to intranuclear chromatin binding of Nups. In this review, we take an expanded examination of research on the relationship between NPC components and chromatin structure, which can manifest at NPCs themselves or at intranuclear locations, and conclude with a survey of some of the recent updates on transcriptional regulation by Nups.

## 2. NPCs, Nups, and Chromatin Structure

### 2.1. Nups and Histone Modifications

With respect to regulation of chromatin state by NPCs/Nups, perhaps the most is known about the interactions of NPCs and their constituent Nups with various histone-modifying enzymes. Histone modifying enzymes are responsible for depositing post-translational modifications on core histone tails. These modifications can either directly regulate chromatin condensation state, as in the case of acetylation, or provide a signal to recruit other proteins that regulate chromatin compaction or downstream transcriptional processes [17]. While the exact molecular function of many histone modifications remains elusive, general associations of specific marks with active or repressed transcription have been well explored, and can provide insight into the possible relationships between chromatin processes and Nups.

#### 2.1.1. Nup98 and Methylation of Histone H3 Lysine K4

Transcriptional memory is the phenomenon of enhanced rounds of subsequent transcriptional activation after an initial stimulus-induced transcriptional event, a process previously reviewed extensively [18,19]. Some of the first evidence of the association between Nups and active chromatin marks came from studies of human Nup98 and its yeast homolog in regulation of transcriptional memory. In these experiments, enhanced reactivation or memory of inducible genes required Nup98 and the deposition of a specific histone modification H3K4me2 [14], a mark associated predominantly with promoters of active genes [20,21]. During establishment of transcriptional memory, H3K4me2 deposition at promoters is dependent on Nup98, and this is true in both yeast and human cells [14]. The histone methyltransferase (HMT) Set1 (capable of both di- and tri-methylation of H3K4) was shown in yeast to be required for this deposition, thus it is plausible that Nup98 may promote the interaction of Set1 with promoters undergoing transcriptional memory establishment. Interestingly, binding of Nup98, which is known to be highly dynamic [16], to such promoters appears to predominantly occur in the nuclear interior in human cells. On the other hand, in yeast, genes regulated by the Nup98 homologue Nup100 localize predominantly to nuclear pores at the nuclear periphery upon activation and transcriptional memory establishment. It thus appears likely that the interaction between Nup98 and Set1 can occur regardless of subnuclear location.

The relationship between Nup98 and H3K4 HMTs is further supported by data in human hematopoietic progenitor cells, where Nup98 interacts with a component of the homologous HMT Set1A/COMPASS complex and is required for the targeting of the complex to promoters [22]. Genome-wide, chromatin binding of Nup98 in these cells is often adjacent to a related histone mark H3K4me3, deposited by Set1A/COMPASS and associated with active promoters, and depletion of Nup98 results in decreased deposition of H3K4me3 at co-targeted promoters [22]. In *Drosophila*, Nup98 was found to similarly interact with a homologous H3K4 HMT, Trithorax (Trx), both physically and genetically, and to regulate expression of Trx target genes [9]. This is especially interesting, because while yeast Set1 and human Set1A are direct homologs, Trx is more distantly related [23]. The evolutionary conservation of these interactions suggests that Nup98 is intimately involved in regulation of specifically di- and trimethylation of H3K4, underscoring an important functional role of Nups in regulation of histone modifications. The precise mechanism for this role of Nup98 remains to be determined, but as discussed above, one likely possibility is that Nup98 facilitates recruitment of Set1 or Trx HMTs to chromatin or promotes stabilization of their chromatin binding. This potential role of Nup98 in stabilization of a particular chromatin state may also be linked to transcriptional memory-associated chromatin looping [24], which has been recently reviewed extensively elsewhere [25].

#### 2.1.2. Nups and Histone Acetylation

Another NPC component Nup153 has similarly been shown to interact with histone modifying enzymes associated with the active chromatin state. Early ChIP-chip experiments in *Drosophila* revealed binding of Nup153 predominantly at active loci, marked by RNA polymerase II (pol II) [26]. More recently, a relationship between Nup153 and CBP/p300 has been observed in mammalian cells. CBP/p300 are part of a well-studied histone acetyl transferase (HAT) complex known to acetylate both histones and non-histone proteins [27]. The chromatin-related activity of CBP/p300 is robustly associated with chromatin decondensation and gene activation [28], which is most likely related to its role in histone acetylation. The HAT activities of CBP and p300 are known to target multiple histone residues, but one of its best-known targets is acetylation of H3K27, a mark associated with active genes and particularly with active enhancers [29]. Generally, histone acetylation has been shown to biophysically induce chromatin decompaction and DNA accessibility in vitro and in vivo [17], and is thought to promote gene activity through such decompaction. In cardiac tissue, Nup153 was found to interact with p300 and p300/CBP associated factor (PCAF) and co-target a similar set of genes, which depended on Nup153 for normal expression [30]. Additionally, Nup153 upregulation resulted in an increase in p300/PCAF acetylation activity, while knock-down of Nup153 lead to a decrease [30]. Together these data suggest that Nup153 plays a role either in recruiting PCAF/p300 to chromatin, or in increasing its acetylation activity once there, to promote expression of cell type specific target genes. Further data in support of these findings can be found in a paper exploring the role of Nup98 FG domains, which are recurrent domains of Phenylalanine and Glycine repeats found in Nups, in human leukemia-associated transcriptional upregulation. This study found that the FG domain of Nup98 physically associates with CBP in vitro and in vivo, and this interaction facilitated Nup98-induced expression of a target reporter gene [31]. Importantly, the same study also reported that the Nup153 FG domain similarly increased reporter gene activation in the same assays, which supports the notion that Nup153 interacts with CBP. In further confirmation of this relationship, ChIP-seq studies in multiple human cell lines demonstrated enrichment of Nup153 at enhancers [32] (discussed further below). Similarly, Nup98 was found to target multiple enhancers in the fly genome [24]. As discussed above, CBP and H3K27 acetylation are highly abundant at enhancers (they are often utilized as genomic marks to identify enhancers), supporting the notion that Nup153 and Nup98 may interact with CBP in a functionally relevant context. Together, these findings reinforce an evolutionary conserved role of Nups in regulating gene expression through interactions with/recruitment of histone modifying enzymes (Figure 1).

#### 2.1.3. High Resolution Imaging of Chromatin at Nuclear Pores

In an attempt to characterize the chromatin landscape at the nuclear pore, one group has recently utilized high resolution structured illumination microscopy (SIM) and sophisticated data analysis software. Importantly, they were able to confirm that areas of open chromatin, or heterochromatin exclusion zones, at nuclear pores, observed in previous EM images, can be reproduced by DAPI stain and high resolution fluorescence imaging in HeLa cells [33]. In further support of this, they were able to see that heterochromatin-associated histone modifications such as H3K27me2 and H3K9me2 were explicitly excluded from this same region. Perhaps unsurprisingly, this was not true for active chromatin marks H3K4me2 or H3K9ac, which were detected at nuclear pores. Interestingly, the authors also found histone demethylase LSD1 present at the nuclear pore. LSD1 is known to demethylate both H3K4me2/1, commonly associated with active promoters, and H3K9me2/1, commonly associated with condensed repressed heterochromatin. As there is a deficit of this heterochromatin mark, and no deficit of this active chromatin mark observed at nuclear pores, it is enticing to envision that the nuclear pore utilizes LSD1 to help regulate the balance between the repressive and active methylation marks to promote a specific chromatin environment. One possible scenario could be promoting a “ground” demethylated H3K4 state, onto which specific dimethylation marks can be deposited upon a signaling event to establish transcriptional memory, described above. Another possibility is the ability of the nuclear pore to impart specificity on the demethylating activity of LSD1, resulting in preferential removal of H3K9 methylation in NPC-associated chromatin environment. Further research will be required to address these and other possibilities, but the identified connections of the NPC to the machinery that regulates H3K4 methylation, such as Set1, Trx, and LSD1, highlight the functional role of NPC–genome interactions in regulation of chromatin state.

#### 2.1.4. Activation and Repression Dichotomy–Nup155/Nup170p and Chromatin Silencing

One clear theme within the field of nuclear pore proteins, whether in their roles in transport or their roles in regulating chromatin state and gene expression, is that the ~30 Nups comprising the nuclear pore are individual proteins that can have highly unique functions. Although many NPC constituents seem to be involved in regulating transcriptional activation, there are examples where Nups are involved in facilitating formation of repressed chromatin and reducing gene expression levels. One potential explanation for this could be that the Nups bound to condensed chromatin may be promoting the divide or the transition between euchromatin and the adjacent lamina-associated heterochromatin. Another could be that not all nuclear pores in a nucleus perform the same functions, and that a sub-population of nuclear pores may be involved in gene repressive functions. While the reasoning behind this dichotomy is still a mystery, it is becoming increasingly clear that specific Nups have a functional relationship with silenced genes (Figure 1).

One of the first studies on this relationship showed an interaction between Nup155 and Histone Deacetylase 4 (HDAC4) in human cardiomyocytes [34]. HDAC4 is a histone deacetylase (HDAC), canonically implicated in inducing chromatin compaction and gene repression [17,35]. Nup155, a core component of the NPC Nup93 sub-complex, was found to physically interact with HDAC4 in these cells [34]. When Nup155–HDAC4 interaction was inhibited, HDAC4 target gene localization to the periphery was generally reduced and the expression of many HDAC4 target cardiac genes increased, indicating that Nup155 normally promotes HDAC4′s silencing capabilities, and does so at peripheral NPCs [34]. Strikingly, Nup155′s role in promoting chromatin compaction and repression appears to be highly conserved. The yeast homolog of Nup155, Nup170p, was found to be required for localization of the silencing factor Sir4 to subtelomeric chromatin [36]. While not a histone modifier itself, Sir4 is well-established as a critical protein required for chromatin compaction in yeast, recruiting histone deacetylases, which in turn promote formation of repressive chromatin [37]. The functional effect of Nup170p on chromatin silencing was supported by its observed chromatin binding to subtelomeric chromatin, supporting the notion that this is a direct effect of the chromatin binding of Nup170p as opposed to possible indirect transport effects [36]. Although it was not directly tested, this phenomenon is assumed to also occur at NPCs at the nuclear periphery, as Nups of the core Nup93 and Nup107 sub-complexes are known to be stably associated with the NPC [16], and most of the interactions between yeast Nups and the genome have been found at the nuclear periphery [38]. A few years later, it was shown that the interactions between Nup170p and Sir4 appear to exist within a complex with a subset of Nups and telomere-localizing machinery, and distinct from fully intact NPCs in the nuclear envelope [39]. These findings lend credence to the hypothesis that there may be different NPCs or sub-complexes of the NPC, consisting of a subset of Nups and distinct from the holo-NPC complex, in the nuclear envelope with unique functions, an intriguing concept worthy of further study. Compellingly, the importance of Nup170p role in facilitating chromatin compaction is bolstered by the finding that it also utilizes chromatin remodeling proteins in addition to histone modifiers in promoting repression (discussed further in the next section).

#### 2.1.5. Nups and Polycomb Repression

Interestingly, not only can different Nups have opposing roles in regulation of chromatin state, but it appears that even the same Nup can have opposing functions depending on cell type or developmental context. Nup153, which as we previous discussed, associates with the CBP/p300 complex to promote gene expression in cardiac tissue, has also been shown to carry out a gene repressive role in mouse Embryonic Stem Cells (mESCs) [40]. In this study, they found that loss of Nup153 in mESCs resulted in de-repression of many developmental genes, and thus promoted early differentiation. The de-repressive effect of Nup153 was consistent with its chromatin binding profile, obtained by DamID mapping, which revealed co-localization of Nup153 with Polycomb Repressive Complex 1 (PRC1) components at Transcription Start Sites (TSSs) of a subset of repressed differentiation genes. Furthermore, Nup153 was found to biochemically interact with PRC1 components, and its loss reduced binding of the PRC1 component Ring1 to target differentiation genes [40]. In this case, target genes of Nup153 were identified both at NPCs at the nuclear periphery and in the nuclear interior, suggesting that the Nup153–Polycomb interaction is independent of nuclear location. PRC1 is known to co-function with the PRC2 complex, which carries a histone-modifying component responsible for deposition of the repressive mark H3K27me3. Importantly, Nup153, along with Nup107 and Nup62, were also found to regulate occupancy and activity of other Polycomb complex components and regulate repression of imprinted genes in mouse embryonic endoderm cells [41]. Consistently, another NPC component, Nup93 has been implicated in peripheral tethering and repression of the *HoxA* gene, where it was found to promote the Polycomb-associated mark H3K27me3 [42]. The mechanism behind how Nup153 or other Nups can function both in the recruitment of the repressive Polycomb proteins, and in the recruitment of the activating CBP/p300 complex, at different genes or in distinct cell types, is a highly intriguing question for future research. Regardless, the importance of Nups in promoting recruitment of histone-modifying proteins to affect chromatin structure and downstream gene expression has now been well documented, and is an exciting notion considering these proteins were once thought to exclusively function in nucleocytoplasmic transport.

**Figure 1 cells-08-01414-f001:**
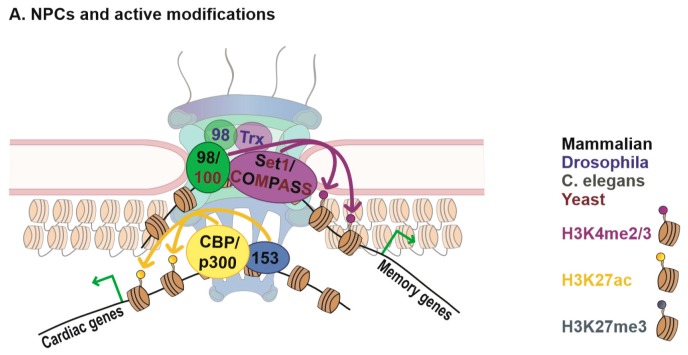
Nuclear pore complexes (NPCs) and Regulation of Chromatin States. (**A**) NPCs and active histone modifications. Mammalian and yeast Nup98/Nup100 have been shown to regulate binding of HMT complex Set1/COMPASS to chromatin, and influence deposition of downstream active chromatin marks H3K4me2/3 [14,22]. Association of *Drosophila* Nup98 and the related H3K4me2 HMT Trx have also been demonstrated [43], and in all cases, Nup98 or its homologs were required for target gene activity. Additionally, mammalian Nup153 has been found to interact with and regulate histone acetyl transferase (HAT) activity of the CBP/p300 complex at target genes involved in cardiac development in cardiac tissue, subsequently promoting gene expression [30]. Note that these Nup–chromatin interactions have been described at both peripheral NPCs and in the nuclear interior, although only the NPC location is shown here. (**B**) NPCs and repressive histone modifications. Nup155 has been shown to physically interact with HDAC4 in mammalian cardiomyocytes, and promote HDAC4 binding to, and silencing of, target genes [34]. A biochemical interaction between Polycomb repressive complex PRC1 and Nup153 has been reported, with Nup153 facilitating chromatin binding of PRC1 and downstream H3K27me3 modification and repression of target genes [40,41]. Another nucleoporin, Nup93, has been shown to regulate H3K27me3 occupancy of developmental gene *HoxA* in mammalian cells [42]. While both Nup155 and Nup93, being core NPC components, are thought to interact with repressive chromatin primarily at the nuclear periphery, Polycomb-bound targets of Nup153 have also been detected intranuclearly. (**C**) NPCs and chromatin remodeling. Yeast Nup170p binds repressed subtelomeric and ribosomal protein genes, physically interacts with chromatin remodeler RSC, and is important for maintaining high nucleosome occupancy at target repressed genes [36]. The *Drosophila* nucleoporin Elys has been found to interact with the chromatin remodeling complex PBAP and function to reduce nucleosome occupancy of target genes [44], while interaction between the *Caenorhabditis elegans* (*C. elegans)* homologs Mel-28 and swsn-2.2 has also been observed [45]. Currently, the nuclear location of Elys–chromatin interactions has not been determined, and may occur intranuclearly as well as at the NPC.

### 2.2. Nups and Chromatin Remodeling

Chromatin remodeling proteins regulate chromatin structure in more direct ways than histone modifiers. While histone modifying enzymes deposit marks on histone tails that can regulate nucleosomal interactions or provide binding platforms for downstream proteins, chromatin remodelers possess an intrinsic enzymatic activity to alter physical interactions between histones and DNA, changing the occupancy or the spacing of nucleosomes. This activity can function through eviction of one or more of the core histones within the histone octamer, a sliding of the histone octamer down the DNA sequence, or a complete removal of the entire octamer to generate what is referred to as a Nucleosome Free Region (NFR) [46]. These regions are considered highly important in facilitating access of transcription factors and transcriptional machinery to their target sequences in gene promoters and enhancers (reviewed extensively in [47,48]). Below we discuss several examples, in which Nups interact with, and/or affect recruitment or function of chromatin remodeling proteins, providing evidence that a key role of specific Nups is to regulate chromatin compaction and accessibility for downstream transcription (Figure 1).

#### 2.2.1. Nup170p and RSC

As described in the previous section, mammalian Nup155 interacts with repressive histone modifiers to facilitate chromatin compaction and target gene repression. Interestingly, its yeast homolog, Nup170p, is also involved in facilitating increased chromatin condensation and repression, but instead appears to employ chromatin remodeling proteins, such as RSC, a chromatin remodeling complex involved among other functions, in telomere maintenance [36]. A genetic screen for Nup170p interactions yielded several protein complexes associated with formation of repressive chromatin, including a chromatin remodeler SWR1, HDACs Rpd3L and Set3C, and a histone H2BK123 ubiquitylase. However, a physical interaction with these specific proteins could not be identified, suggesting that the genetic interaction was due to functioning in similar pathways. When analyzing differentially expressed genes in a Nup170p mutant line, 90% showed an increase in expression, demonstrating a functional role of Nup170p in global gene silencing. Among the upregulated genes, many ribosomal protein (RP) and subtelomeric genes were found, which led to discovery of a physical interaction between Nup170p and Sth1p of the RSC chromatin remodeling complex, associated with telomere maintenance and repression of subtelomeric genes. Consistently, Nup170p was found at many of these subtelomeric genes by ChIP, and loss of Nup170p resulted in a decrease in nucleosome occupancy surrounding subtelomeric gene TSSs, which pheno-copied Sth1p depletion [36]. Together these data strongly imply a functional role for Nup170p in the recruitment and/or activity of the RSC remodeling complex at these genetic targets, resulting in more condensed chromatin and downstream transcriptional repression.

#### 2.2.2. Elys and Chromatin Remodeling

Multiple studies have now demonstrated the role for an NPC component Elys and its homologs in seeding post-mitotic nuclear pore formation by recruitment of the core Nup107 sub-complex to chromatin [49,50]. Consistent with this, Elys is the only Nup with a putative DNA binding domain and a demonstrated nucleosome binding capacity in vitro [49,51,52,53]. Interestingly, however, Elys was originally discovered as a Transcription Factor (TF) capable of inducing expression when targeted to a reporter gene [54]. The mechanism of this transcriptional function of Elys has not been elucidated, yet there is evidence to suggest it may be through interactions with chromatin remodelers and possible chromatin accessibility increase. A recent study from our lab has identified an interaction between Elys and the PBAP chromatin remodeling complex, and found that reduction of Elys levels increases target gene nucleosome occupancy in *Drosophila* [44]. This function of Elys is supported by several previous reports of Elys interacting with chromatin remodeling proteins throughout evolution. The *C. elegans* homolog of Elys, Mel-28, has been seen to interact with an accessory subunit of the SWI/SNF chromatin remodeling complex, which is homologous to the *Drosophila* PBAP complex. Specifically, Mel-28 was found to interact with the component swsn-2.2 [45], which is a homolog of the human BAF-60, a remodeling complex component important for decondensing chromatin to allow for downstream transcriptional processes [55]. A genetic interaction screen in the background of Mel-28 RNAi revealed an interaction with *pyp-1*, the homolog of another *Drosophila* chromatin remodeling protein NURF-38, resulting in larval sterility [56]. NURF-38 is a core component of the Nucleosome Remodeling Factor (NURF) chromatin remodeling complex and promotes chromatin accessibility for downstream transcription [57,58]. In human cells, a mass spectrometry screen unearthed an interaction between Elys and HMGN3 [59], a member of a family of proteins known to interact with histone H1, and to promote chromatin decondensation and upregulation of target genes [60]. This evolutionarily conserved relationship between Elys homologues and chromatin remodelers may help provide a possible mechanism for the long-observed zones of heterochromatin exclusion at NPCs at the nuclear periphery, and offer another mechanism by which NPCs/Nups are capable of regulating transcriptional states.

Consistent with the reported interactions between Elys and chromatin remodeling proteins, several recent studies revealed a functional role of Elys in regulating global chromatin compaction. One study looking at DNA replication in *C. elegans* found a defect in global genome decondensation in the background of mutant replication machinery, but this phenotype was rescued in an Elys mutant, suggesting that Elys may be involved in the crosstalk between replication and chromatin compaction although the precise molecular role of Elys in this process remains unclear [61]. In a subsequent investigation, the authors examined genomic decondensation that occurs upon fertilization in the *Xenopus* sperm nucleus. Here they found that treatment with RNases depleted Elys off of chromatin, and this resulted again in defects in chromatin decondensation [62]. Specifically, they observed smaller nuclei with more condensed chromatin, as judged by nuclear H2B immunofluorescence stain and EM, and elevated resistance to MNase digestion. The increased chromatin compaction was also shown to not be due to transport defects, as treatment with Wheat germ agglutinin (WGA), which blocks nucleocytoplasmic transport, did not result in this phenotype. Finally, the study from our lab demonstrating a role for Elys in gene-specific chromatin decondensation also found reduction of Elys to cause a global defect in genome decondensation in *Drosophila* cells [44]. This data combined with the previously discussed relationships between Elys and various chromatin remodelers suggest a critical role for Elys in regulation of chromatin decompaction through interaction and regulation of chromatin remodeling proteins. Interestingly, another Nup, Seh1, which is a component of the Nup107 sub-complex associated with Elys, was similarly found to regulate chromatin accessibility of target genes involved in differentiation, as measured by ATAC-seq, in mammalian oligodendrocyte progenitor cells [63].

### 2.3. Nups and Large-Scale Chromatin Structure

Above, we have discussed the current knowledge on the interactions of Nups with specific histone-modifying and chromatin remodeling enzymes, as well as cases where Nups have been demonstrated to functionally contribute to deposition of specific histone modifications or to chromatin compaction. In the section below, we examine instances of known functions of Nups in regulation of large-scale chromatin and nuclear organization. Mechanistically, these roles may be similarly based on the known interactions of Nups with chromatin regulatory machinery, or they may involve a unique set of molecular mechanisms. Either way, in the cases presented below, these mechanisms are manifested in chromosome-wide epigenetic processes, in structural elements of the nucleus, and in genome integrity.

#### 2.3.1. Nups in Dosage Compensation

Over recent years, several papers have described the role of Nups in dosage compensation (DC). The unequal number of X chromosomes between male and female organisms requires a process, via which gene expression from the X chromosome is equalized between the sexes, and which different species accomplish utilizing different mechanisms. This level of transcriptional regulation is no small feat, as it requires changing the transcriptional output of an entire chromosome. In *Drosophila* the Dosage Compensation Complex (DCC) binds along the length of the X-chromosome, as is a common theme for DC machinery in all species and is responsible for ~2-fold upregulation of X-linked genes in male flies. The DCC contains a HAT called Males absent On the First (MOF) responsible for depositing the activating histone mark H4K16acetyl, which coats the male X-chromosome and is necessary for transcriptional upregulation [64]. In male *Drosophila* embryos, Nups, Mtor, and Nup153 were found to interact with MOF and other DCC components biochemically, as both Nups were recovered as abundant hits in the proteomic characterization of the interacting partners of MOF [65]. Upon Nup depletion, the normally robust localization of MOF and other DCC machinery to the male X chromosome appeared to be compromised, along with the downstream transcriptional upregulation of certain X-linked genes [65]. Interestingly, the authors also found a physical interaction between the human orthologs of Mtor (TPR) and MOF, even though human cells do not employ hMOF in their DC process of inactivating one of the female X chromosomes. In line with these findings, chromatin binding patterns of Mtor and Nup153, assessed by ChIP-chip, were found throughout the *Drosophila* genome in large domains termed Nucleoporin Associated Regions (NARs), enriched especially at transcriptionally active regions [26]. Fascinatingly, ~70% of these domains were localized to the male X-chromosome, and were enriched for. Furthermore, this study demonstrated a robust co-localization of the genome-wide binding patterns of the Nups with H4K16acetyl and MOF [26]. In another intriguing conservation of function, interactions between the X chromosome, dosage compensation, and nuclear pores have been found in *C. elegans* as well, even though DC is accomplished by condensation and down-regulation of the X, rather than activation and transcriptional upregulation [66]. One comment of note is that the importance of Nups in *Drosophila* dosage compensation has been questioned. A more recent study did not observe an effect of Nup153 or Mtor on X-chromosome expression or recruitment of the MOF complex in *Drosophila* larval tissues or S2 cells, and the authors proposed a difference in knock-down methods as a potential culprit [67]. However, the reported evolutionary conservation of protein–protein interactions between Nups and DCC machinery in *Drosophila* and humans [65], the interaction of nuclear pores and the male X-chromosome in *C. elegans* [66], and the discovery of large binding domains of Nups on the *Drosophila* male X-chromosome [26], all support the hypothesis that certain Nups have a conserved function in DC, and suggest that this topic warrants further study.

#### 2.3.2. Mtor/TPR and Nuclear Organization

Another recent study provided an example of the function of nucleoporin Mtor in large-scale genome organization. Here they discovered that Mad1, a protein normally part of the mitotic spindle assembly checkpoint, was found in post-mitotic and interphase *Drosophila* spermatocyte nuclei in what the authors termed Mad1-containing IntraNuclear Territories (MINTs) [68]. Of particular interest, other proteins identified in the chromatin-associated MINTs included Mtor, which canonically has been shown to anchor Mad1 to the nuclear envelope for its spindle associated functions in both *Drosophila* and human cells [69,70]. Furthermore, Mtor was required for the formation/maintenance of these intranuclear bodies, as depletion of Mtor caused a complete dissolution of proteins associated with MINTs. The function of MINTs is so far unknown, but one hint may be in the chromatin-associated role these authors found for Mad1. To test if Mad1′s localization to these intranuclear bodies had anything to do with regulating chromatin function, the authors conducted assays to test for genetic interactions with heterochromatin formation and with Polycomb repression, and found that in both cases the endogenous function of Mad1 appears to promote open chromatin formation or maintenance [68]. This finding suggests that the function of the intranuclear MINT bodies may involve regulation of open chromatin, and since the existence of these bodies is Mtor-dependent, further implicates a possible role for Mtor in this process.

The described relationship between Mtor and MINTs, and Mtor’s potential role in DC, support the idea that Mtor plays an evolutionarily conserved role in promoting an active chromatin environment. In further support of this idea, an intriguing recent work has implicated Mtor in the distribution of open chromatin along the nuclear periphery. As mentioned at the beginning of this review, the peripheral chromatin localized to NPCs appears open and euchromatic in nature, relative to the adjacent condensed lamina-associated heterochromatin. One group set out to understand what regulates these heterochromatin exclusion zones (HEZs), and found that the mammalian homolog of Mtor, TPR, was required for their formation [71]. In cells in which this Nup was depleted, HEZs were abolished and the heterochromatin at the periphery continued un-disrupted across NPCs. Importantly, these observations were made in mammalian cells, infected with a poliovirus, and it remains unclear whether loss of Mtor/TPR results in a similar loss of nuclear pore-associated open chromatin in any normal cell type. However, HEZs have been detected for decades in many different, wild-type cells across species [1,72], and Mtor appears to play a role in regulating this chromatin distribution in at least certain cellular contexts.

Consistent with the finding that Mtor is important to maintain open chromatin at NPCs, there are multiple studies linking both Mtor and Nup153 with viral genome insertion. One study, using Fluorescent In-Situ Hybridization (FISH) and molecular assays, showed that genomic integration sites for the HIV-1 viral DNA occur predominantly at active chromatin associated with NPCs, as opposed to peripheral lamin-associated heterochromatin, and that Nup153 was required for this localization [73]. An additional study found an involvement of Mtor/Tpr as well, demonstrating that HIV genomic integration into active regions is dependent on Tpr, and overall infectivity is reduced as a result of Tpr depletion [74]. These studies support the notion that the open chromatin environment at NPC is facilitated by Nups, and are in line with previous data supporting this hypothesis, including the requirement of Mtor for HEZs [71].

#### 2.3.3. Nups and Transposon Silencing

Multiple studies have recently demonstrated a role for NPCs and Nups in silencing of transposable elements (TEs). Piwi, a critical component of the piRNA pathway responsible for silencing TEs, uses complementary piRNAs to seek and target TEs for degradation. Interestingly, genomic targets of Piwi were found to substantially overlap with genomic binding maps of NPCs [75]. A further study demonstrated a direct interaction between Nup358 and Piwi [76]. What is especially interesting is the requirement of Nup358 in piRNA biogenesis as well, which is a process not known to rely upon Piwi but on other components of the transposon silencing pathway. Likely due to a combinatorial affect, Nup358 reduction resulted in a de-silencing of TEs, which had a predictable negative effect on genomic stability. As several Nups have been discovered in screens identifying factors involved in TEs silencing, it seems this may be a general function of NPCs/Nups that will likely warrant further study to fully understand [77,78]. The suppression of transposable element expression by NPCs is especially interesting in the context of their other role in protecting genome stability through facilitating DNA damage repair and telomere maintenance (extensively reviewed in [8,79]).

## 3. NPCs and Transcription

Many of the interactions between Nups and chromatin factors, described above, have been shown to fulfill functions in regulation of downstream gene expression. Regulation of gene expression overall has become a well-established function of NPCs and individual Nups. We discuss some of the most recent findings on this topic below, where, in the examples to follow, functional roles of Nups have been linked more directly to transcriptional activation, as opposed to regulation of chromatin structure. In some cases, certain Nups have been shown to play a role in recruitment of specific transcription factors (TFs) or influencing transcription itself, but since regulation of transcription and chromatin structure are intimately intertwined, Nups may affect the transcriptional process via both interactions with TFs and with chromatin machinery. We hope that the above summary of the chromatin-related roles of NPC components will help draw attention to the clear importance of Nups in regulation of chromatin itself, and that these findings may be kept in mind while examining future research as the field of Nups in regulation of gene expression continues to be explored. Early work in this field has established that Nups bind chromatin in multiple cell types, that many of the targets tend to be cell cycle and developmental genes, and that this binding can have an effect on gene expression [11,12,80,81,82]. Over the years, more examples of and mechanisms behind some of these specific phenomena have been further elucidated. Some excellent and extensive reviews on this topic have been written over the years, including [5,7,9], so for detailed information and visual summaries on transcriptional roles of the NPC, we refer the reader to such reviews. Here, we will focus on some of the more recent discoveries in this field and trends among them.

### 3.1. Nup98 and Hox Genes

The theme of Nups binding and regulating cell identity genes has proven robust between cell types and among species. This is particularly intriguing considering the existence of many tissue-specific defects and diseases resulting from Nup mutations and dysfunctions [83,84]. Perhaps the most famous of these involves the Nup98 gene, which is a recurrent member of chromosomal translocations that result in mammalian leukemogenesis [85]. The mechanisms behind Nup98-based leukemias have converged on the chromatin-binding function of Nup98. One recent report found a fusion of the N-terminus of Nup98 with the C-terminus of TF HoxA9 to target many developmental *Hox* genes, which are known to regulate organismal body morphogenesis and cell identity across cell types and organisms [86]. This binding of the Nup98-HoxA9 fusion resulted in upregulation of its targets that are normally repressed, which is a common hallmark of Nup98 fusions. While chromatin targeting of Nup98 fusion proteins is often proposed to be regulated by the Nup98′s fusion partner, endogenous full-length Nup98 has been shown in *Drosophila* to bind and regulate expression of *Hox* genes [43]. This targeting relies on MBD-R2, a TF and a component of the *Drosophila* MOF-containing NSL complex responsible for depositing active histone mark H4K16acetyl on autosomes. Nup98 was proposed to work in conjunction with the canonical *Hox* regulator HMT Trx, as reduction of Nup98 was sufficient to reduce expression of Trx targets and Nup98 was found to physically interact with the H3K4 HMT Trx, as discussed above. Interestingly, mammalian Nup98 was similarly found to physically interact with the homologous NSL and MLL complexes in leukemic models [87]. MLL is a direct homologue of Trx, and translocations and mutations of its gene are another common set of aberrations in human leukemia. Both MLL-based and Nup98-based leukemias are associated with pathogenic upregulation of *Hox* genes, supporting the notion that both MLL and Nup98 normally regulate expression of *Hox* genes. Overall these studies highlight the role of Nup98 in regulating expression of key developmental *Hox* genes in both endogenous and disease contexts. Though the mechanism of this regulation is still being explored, Nup98 in other contexts has been shown to associate with architectural proteins and regulate enhancer-promoter looping, as part of transcriptional memory [24]. It is possible that all of these mechanisms, including H3K4 methylation, recruitment of TFs, and enhancer-promoter looping, underlie the function of Nup98 in regulation of *Hox* genes and other targets. In this manner, Nup98 may primarily function in stabilizing a particular transcriptional complex, an idea discussed recently [25].

### 3.2. Nup153, Nup93, and Regulation of Cell Identity

Multiple reports have recently demonstrated the binding of NPC components to tissue-specific enhancers (reviewed in [25]). In addition to targeting of Nup98 to *Drosophila* enhancers, described above, Nup153 and Nup93 have recently been found to bind super-enhancers in human cells, interestingly predominantly at the nuclear periphery [32], even though Nup153 is mobile relative to NPCs [15,16]. Super-enhancers are described as large clustered enhancers, marked by especially high levels of H3K27 acetylation and presence of binding sites for multiple TFs, and they are especially known for regulation of cell-identity genes [88]. Strikingly, a third or more of super-enhancers were found to have one or both Nups bound in the multiple human cell types analyzed, and the binding of Nup153 or Nup93 was critical for appropriate gene expression [32]. Of special interest, the direction of change in gene expression was not uniform, in that roughly half of each of their gene targets went up and the other half went down upon reduction of either Nup. Indicative of the dichotomy of function described above, this is another interesting example of individual Nups having both positive and negative effects on gene expression in the same cell populations, and is consistent with findings of both Nup153 and Nup93 associated with repressed genes [40,89]. While the possibility of secondary downstream affects in the RNA-seq data should be considered, this could present an interesting case involving the context-dependent nature of Nup functions at different genetic targets and would warrant further study to determine specific mechanisms utilized.

### 3.3. Nups and Transcription Factors

One mechanism behind specificity of function for individual Nups at different genetic targets may lie in differential protein binding partners, especially cell-type/context dependent transcription factors. Such binding partners have been identified for Nup153 in regulation of differentiation in neural progenitor cells (NeuPCs) [90]. Specifically, Nup153 has been found to interact with, and regulate genomic binding of, TF Sox2. While important for maintaining embryonic stem cell pluripotency, Sox2 has also been shown to cooperate with canonical NeuPC transcription factors to regulate maintenance and differentiation of NeuPCs. Accordingly, many of the genes disrupted by Nup153 reduction were associated with neural development, and Nup153 loss promoted differentiation. In a manner consistent with previously discussed negative transcriptional regulation by Nup153 [40], there were an equal number of up- and down-regulated gene targets upon Nup153 reduction, suggesting perhaps the ability to control transcription via multiple mechanisms. Importantly, some of this regulation appears to be through targeting or maintenance of Sox2 on chromatin, as loss of Nup153 reduced Sox2 signal at over half of its genomic targets. Interestingly, the direction of transcriptional regulation by Nup153 correlated with its location on gene targets, in that 5′ localization trended towards facilitating transcription, and 3′ targets were more often associated with gene repression. By comparing their data with microarray expression analysis in the same cell type upon Sox2 shRNA treatment [91], they observed that Nup153 localization within gene targets was distinct from that of Sox2, which was primarily at 5′ TSSs regardless of its transcriptional effect on its targets [90].

Nups regulating binding or activity of cell-type transcription factors to control transcriptional programs is becoming a common theme in the field. It has also been observed for Nup210, which was shown to recruit muscle TF Mef2C, and its genomic targets, to NPCs in myofiber nuclei to promote expression of genes regulating muscle differentiation [92]. Similarly, Nup Seh1 has recently been shown to recruit oligodendrocyte TFs Olig2 and Brd7 to NPCs to promote development of oligodendrocyte progenitor cells [63]. In further support of this trend, a recent large-scale study in yeast has shown that a considerable fraction of all TFs is capable of tethering gene targets to NPCs in the nuclear periphery [93], which can promote target gene expression in yeast [81,94]. These findings provide further evidence to support the view of NPCs and their constituent Nups as transcriptional hubs, utilizing interactions with context-dependent and cell-type-specific transcription factors to promote developmental transcriptional programs. It is of note that previously Nups have been shown to interact and cooperate with general transcription factors/coactivators such as Mediator [95] and the SAGA complex [96], which can be recruited by specific TFs [97], and perform histone acetyl-transferase activity to promote chromatin decondensation and accessibility. Overall this data demonstrates the ability of Nups to regulate transcription of target genes via multiple mechanisms, one of which includes recruiting or stabilizing binding of cell-type specific transcription factors in order to regulate cell identity. The fact that Nups can be linked to various stages in general and specific transcriptional processes is a testament to the multi-functionality of the NPC and its constituents. However, as discussed earlier, one hypothesis that can explain this multi-functionality envisions Nups being involved in multiple interactions at once, with both TFs and chromatin modifying and remodeling machinery, and in this manner, serving a primarily stabilizing or perhaps connective purpose for a competent transcription-chromatin complex.

## 4. Concluding Remarks and Future Perspectives

Over the years, the field has come to learn the pleiotropic nature of NPCs and their constituent Nups, as multiple mechanisms, by which they regulate genomic integrity, mitosis, nuclear organization, transcription, chromatin state and, of course, transport, were discovered. As we have discussed, these proteins have proven to be critical for many roles promoting proper nuclear organization and function, especially with regard to chromatin structure and gene expression. Recently, we have grown to appreciate their importance in cell-type specific gene expression, with certain Nups regulating recruitment of specific transcription factors to their target genes to facilitate developmental transcriptional programs [63,90,92]. We have also highlighted that their role in regulation of transcription may often be carried out through regulation of chromatin state, whether through interactions with histone modification enzymes [14,22,30,40,43,65], interactions with chromatin remodeling enzymes [36,44], or through as-of-yet unknown mechanisms. Based on the involvement of Nups in many tissue-specific phenotypes [72,83,98], understanding their roles in these gene regulatory processes may help answer questions in basic cell biology as well as in human disease.

There are many remaining questions in regard to the precise functional role of NPC components in regulation of chromatin and transcription. As presented above, different Nups appear to be involved in unique molecular mechanisms, and it will be critical to characterize tissue-specific chromatin binding patterns, interacting partners, and organismal phenotypes of individual Nups in order to fully define the NPC–genome relationship. The apparent ability of certain Nups to promote both gene activation and gene repression, depending on the specific genomic and developmental context, is a particularly interesting question for future study. Comprehending this specificity and the Nup–chromatin relationship in general hinges upon determining the molecular basis of Nup recruitment to chromatin, which remains poorly understood. In the majority of cases, it is unclear how Nups, which lack any obvious DNA or chromatin binding domains, are recruited to their target loci, and how their recruitment is influenced by cell type context or chromatin state. It is similarly unclear whether there is a functional difference between Nup–chromatin contacts of different sub-nuclear locations, at peripheral NPCs versus in the nuclear interior, and whether dynamic Nups can re-localize the chromatin template on and off the NPC. It has been suggested that certain loci contact the nuclear pore during activation and re-localize back to the nuclear interior, once fully activated [13]. However, in other contexts, Nup-targeted genes were found at NPCs preceding and following transcriptional activation, suggesting the nuclear pores may function as stable binding scaffolds for certain genes, perhaps to facilitate regulation or to couple to transport [24]. Finally, the relationship between chromatin binding versus transport roles of Nups will need to be characterized. Whether certain genes are recruited to NPCs to couple transcription and transport, as originally suggested by the gene-gating hypothesis [4], or transport activities are fully separated from chromatin binding activities of Nups via the use of different nuclear pores, are key questions to pursue. The latter possibility would suggest a potential competition between gene regulatory and transport activities for NPC components, which is an intriguing idea for further exploration. Future work will be able to address these and other possibilities, shedding light on how the multi-functional NPC regulates key nuclear processes and gene expression patterns.

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
