# Peer review of "Nuclear Pore Proteins in Regulation of Chromatin State"

_cells, 2019, doi:10.3390/cells8111414_

Round 1

Reviewer 1 Report

Kuhn and Kapleson have provided a well written and organized review that outlines known roles for Nups/NPCs in regulating chromatin state. This will undoubtedly serve as an excellent resource to those interested in this topic generally, as well as those interested in specific questions surrounding the chromatin-associated functions of Nups. The major issue I would like to see addressed prior to publication is a discussion around the functions of Nups at vs. away from NPCs (e.g. in the nucleoplasm). It is well documented that various Nups are mobile and exist outside NPCs, making it possible that some functions, including opposing functions, could result from these different populations of Nups. For example, a Nup central to many of the discussions in the review, Nup98, is known to reside in intranuclear GLFG bodies. Yet, this is not discussed in the text and the data is presented in such a way that the reader is left with the impression that all Nup associated functions with respect to chromatin happen at the NE and NPCs. This is also exemplified in Figure 1, which shows these interactions occurring in the context of the pore. A discussion of these data and concepts is needed to include this possibility and/or to argue why this is unlikely to be the case. Depending on the authors approach to this, these changes may also require Figure 1 to be updated.

Minor things of note:

If desired by the authors, I would be interested in some discussion in review about the following. With NPCs functioning as hubs to support chromatin state and transcription, how do the authors envision competition for these interactions between different gene loci and chromatin domains? The numbers of NPCs are limited, and also need to function to support transport, as such, could there be competition for NPCs? Is this likely to play a role in gene expression regulation etc.?

Line 266, “DNA bicatnding” should be DNA binding.

The final sentence starting on line 526 is rather complex, but the point is critical to the review, hence I suggest edits to this for clarity.

Author Response

Reviewer 1.

Kuhn and Kapleson have provided a well written and organized review that outlines known roles for Nups/NPCs in regulating chromatin state. This will undoubtedly serve as an excellent resource to those interested in this topic generally, as well as those interested in specific questions surrounding the chromatin-associated functions of Nups. The major issue I would like to see addressed prior to publication is a discussion around the functions of Nups at vs. away from NPCs (e.g. in the nucleoplasm). It is well documented that various Nups are mobile and exist outside NPCs, making it possible that some functions, including opposing functions, could result from these different populations of Nups. For example, a Nup central to many of the discussions in the review, Nup98, is known to reside in intranuclear GLFG bodies. Yet, this is not discussed in the text and the data is presented in such a way that the reader is left with the impression that all Nup associated functions with respect to chromatin happen at the NE and NPCs. This is also exemplified in Figure 1, which shows these interactions occurring in the context of the pore. A discussion of these data and concepts is needed to include this possibility and/or to argue why this is unlikely to be the case. Depending on the authors approach to this, these changes may also require Figure 1 to be updated.

We thank the reviewer for their positive assessment and helpful suggestions. We agree that this is an important point to mention and discuss, and have now added documentation and discussion of the nucleoplasmic roles of Nups in chromatin regulation into multiple places in the manuscript, including Introduction, Sections 2 and 3, and Concluding Remarks and Future Perspectives. In fact, we bring up the questions surrounding this topic as one of the major unanswered questions in the field, in the last paragraph. Although we did not change the actual figure, we have added additional information into figure legend, explicitly stating what is known about the nuclear location of depicted interactions.

Minor things of note:

- If desired by the authors, I would be interested in some discussion in review about the following. With NPCs functioning as hubs to support chromatin state and transcription, how do the authors envision competition for these interactions between different gene loci and chromatin domains? The numbers of NPCs are limited, and also need to function to support transport, as such, could there be competition for NPCs? Is this likely to play a role in gene expression regulation etc.?

We thank the reviewer for bringing up this interesting point, and we have added a brief discussion of it into the last paragraph of the manuscript, under Concluding Remarks and Future Perspectives.

- Line 266, “DNA bicatnding” should be DNA binding.

Corrected.

The final sentence starting on line 526 is rather complex, but the point is critical to the review, hence I suggest edits to this for clarity. 

We apologize but we were not sure if the reviewer is referring to the final sentence of the manuscript, or a sentence that encompasses line 526 (in our version, no sentence is starting at line 526), but to address the reviewer’s comments, we tried to clarify both.

Reviewer 2 Report

Kuhn and Capelson wrote a review highlighting the role of nuclear pore complexes (NPCs) and their constituent nucleoporins (Nups) in chromatin structure, genome maintenance, and transcription. I enjoyed reading this nice review. All I have are minor points to consider before its publication.

Minor points:

Page 2, line 46: ‘DNAse’ may be ‘DNase’. The last sentence of the page 3, lines 138-142 is unclear to me. Are they suggesting that LSD1 reduces H3K4 methylation levels in heterochromatin exclusion zones? Or, are they suggesting that LSD1 increases H3K4 methylation levels via reducing H3K9 methylation levels? Can the authors explain the way the nuclear pore utilizes LSD1 to regulate H3K4 methylation levels? Page 7, line 266: ‘bicatnding’ must be ‘binding’. Page 7, line 277: ‘directly homologous” may be ‘homologous. Page 7, line 266: ‘fount’ must be ‘found’. Page 7, line 295: Can the authors explain more about how Elys and DNA replication may function in the same pathway in global genome decondensation? Page 7, line 301: Can the author spell out ‘WGA’ for non-specialists. Page 7, Line 302: ‘block’ must be ‘blocks’. Page 8, Line 320: ‘, (comma)’ must be ‘. (period)’. Page 8, Line 355: …all support of the hypothesis… delete ‘of’ Can the authors confirm the format of the following papers that appear in the reference list. Casolari et al.; Gottesfeld et al.; Kasper et al.; Kuhn et al.; Labade et al.; Raices et al., Raich et al.; Schneider et al.; Seto et al.

Author Response

Reviewer 2.

Kuhn and Capelson wrote a review highlighting the role of nuclear pore complexes (NPCs) and their constituent nucleoporins (Nups) in chromatin structure, genome maintenance, and transcription. I enjoyed reading this nice review. All I have are minor points to consider before its publication.

We thank the reviewer for their positive, helpful and thorough review.

Minor points:

- Page 2, line 46: ‘DNAse’ may be ‘DNase’.

Corrected.

- The last sentence of the page 3, lines 138-142 is unclear to me. Are they suggesting that LSD1 reduces H3K4 methylation levels in heterochromatin exclusion zones? Or, are they suggesting that LSD1 increases H3K4 methylation levels via reducing H3K9 methylation levels? Can the authors explain the way the nuclear pore utilizes LSD1 to regulate H3K4 methylation levels?

We agree that this statement was not clear, and have included the following explanation to clarify our points (now lines 156-164):

“As there is a deficit of this heterochromatin mark, and no deficit of this active chromatin mark observed at nuclear pores, it is enticing to envision that the nuclear pore utilizes LSD1 to help regulate the balance between the repressive and active methylation marks to promote a specific chromatin environment. One possible scenario could be promoting a “ground” demethylated H3K4 state, onto which specific dimethylation marks can be deposited upon a signaling event to establish transcriptional memory, described above. Another possibility is the ability of the nuclear pore to impart specificity on the demethylating activity of LSD1, resulting in preferential removal of H3K9 methylation in NPC-associated chromatin environment. Further research will be required to address these and other possibilities, but the identified connections of the NPC to the machinery that regulates H3K4 methylation, such as Set1, Trx and LSD1, highlight the functional role of NPC-genome interactions in regulation of chromatin state.”

- Page 7, line 266: ‘bicatnding’ must be ‘binding’.

Corrected.

- Page 7, line 277: ‘directly homologous” may be ‘homologous.

Corrected.

- Page 7, line 266: ‘fount’ must be ‘found’.

Corrected.

- Page 7, line 295: Can the authors explain more about how Elys and DNA replication may function in the same pathway in global genome decondensation?

Since this finding has been primarily described in terms of genetic interactions, the precise role of Elys in the DNA replication-genome decondensation relationship seems unclear. We have now explicitly stated this point and re-worded this section, as follows (now lines 333-337):

One study looking at DNA replication in C. elegans found a defect in global genome decondensation in the background of mutant replication machinery, but this phenotype was rescued in an Elys mutant, suggesting that Elys may be involved in the crosstalk between replication and chromatin compaction although the precise molecular role of Elys in this process remains unclear (Sonneville et al., 2015).”

- Page 7, line 301: Can the author spell out ‘WGA’ for non-specialists.

We spelled out WGA as suggested.

- Page 7, Line 302: ‘block’ must be ‘blocks’.

Corrected.

- Page 8, Line 320: ‘, (comma)’ must be ‘. (period)’.

Corrected.

- Page 8, Line 355: …all support of the hypothesis… delete ‘of’

Corrected.

- Can the authors confirm the format of the following papers that appear in the reference list. Casolari et al.; Gottesfeld et al.; Kasper et al.; Kuhn et al.; Labade et al.; Raices et al., Raich et al.; Schneider et al.; Seto et al. 

Confirmed. We have checked the formatting of the following papers, and have not identified any errors.

Reviewer 3 Report

The manuscript entitled ‘Nuclear Pore Proteins in Regulation of Chromatin State’ by Kuhn and Capelson provides a very comprehensive review about the functional connection between NPCs or Nups and the regulation of Chromatin structure and function. It was a great pleasure the read this manuscript. The image presented is very illustrative.

Therefore, I consider that the premise of this study is very interesting and important for the field and I will perform some comments and suggestions.

Minor concerns:

Could the authors summarize the second part of the manuscript (point 3: NPs and transcription) using another image? The manuscript will significantly benefit with it. In the line 172 the authors mentioned that ‘These findings lend the credence to the hypothesis that there may be different NPC or NPC-like structures in the nuclear envelope with distinct functions…’. What exactly are NPC-like structures? Are also Nups? Are they located at the NE? It is not clear to me. I believe that the manuscript would benefit with a section named future perspectives, facing the very promising results presented along the manuscript. There is a mistake in the line 266, I believe you want to spell ‘DNA binding’.

Author Response

Reviewer 3.

The manuscript entitled ‘Nuclear Pore Proteins in Regulation of Chromatin State’ by Kuhn and Capelson provides a very comprehensive review about the functional connection between NPCs or Nups and the regulation of Chromatin structure and function. It was a great pleasure the read this manuscript. The image presented is very illustrative.

Therefore, I consider that the premise of this study is very interesting and important for the field and I will perform some comments and suggestions.

We thank the reviewer for their positive review and insightful suggestions.

Minor concerns:

- Could the authors summarize the second part of the manuscript (point 3: NPs and transcription) using another image? The manuscript will significantly benefit with it.

To address the reviewer’s concerns, we have incorporated the transcriptional aspect of NPC function into the Graphical Abstract of the manuscript. But since our review primarily focuses on chromatin structure, we wanted to focus the figure specifically on these mechanisms. Many other recent reviews have provided detailed overview and beautiful images specifically on the topic of transcriptional regulation by Nups, and we have now explicitly referred the readers to these reviews and their visual summaries of the NPC-transcription relationship on lines 480-481.

- In the line 172 the authors mentioned that ‘These findings lend the credence to the hypothesis that there may be different NPC or NPC-like structures in the nuclear envelope with distinct functions…’. What exactly are NPC-like structures? Are also Nups? Are they located at the NE? It is not clear to me.

We rephrased this statement as follows (now lines 200-202):

“These findings lend credence to the hypothesis that there may be different NPCs or sub-complexes of the NPC, consisting of a subset of Nups and distinct from the holo-NPC complex, in the nuclear envelope with unique functions...”

- I believe that the manuscript would benefit with a section named future perspectives, facing the very promising results presented along the manuscript.

We agree with the reviewer and have added an additional paragraph at the end of the manuscript, discussing future perspectives with a focus on unanswered questions (now in lines 597-623). We have also renamed the last section/section 4 as “Concluding Remarks and Future Perspectives”.

- There is a mistake in the line 266, I believe you want to spell ‘DNA binding’.

Corrected.